# High-Pressure Pasteurization of Oat Okara

**DOI:** 10.3390/foods12224070

**Published:** 2023-11-09

**Authors:** Amanda Helstad, Ali Marefati, Cecilia Ahlström, Marilyn Rayner, Jeanette Purhagen, Karolina Östbring

**Affiliations:** Department of Food Technology Engineering and Nutrition, Lund University, Naturvetarvägen 12, 223 62 Lund, Sweden; amanda.helstad@food.lth.se (A.H.);

**Keywords:** oat okara, high-pressure pasteurization, shelf life, food waste

## Abstract

The issue of the short microbiological shelf life of residues from the plant-based beverage industry creates a large food waste problem. Today, the oat beverage residue, in this study referred to as *oat okara*, is generally converted to energy or used as animal feed. High-pressure pasteurization (200 MPa, 400 MPa, and 600 MPa) was applied to oat okara to investigate the effect on shelf life and microbiological activity. A 4-week microbiological storage study was performed and thermal properties, viscosity, and water and oil holding capacities were analyzed. The total aerobic count, including yeast and mold, was significantly reduced (*p* < 0.05) by 600 MPa after four weeks of storage at 4 °C. The content of lactic acid bacteria after four weeks of storage was low for untreated oat okara (3.2 log CFU/g) but, for 600 MPa, the content remained at the detection limit (2.3 log CFU/g). Conversely, the treatments of 200 MPa and 400 MPa increased the microbial content of the total aerobic count significantly (*p* < 0.05) after two weeks in comparison to untreated oat okara. The thermal properties of untreated and high-pressure-treated oat okara demonstrated an increase in protein denaturation of the 12S globulin, avenalin, when higher pressure was applied (400–600 MPa). This was also confirmed in the viscosity measurements where a viscosity peak for avenalin was only present for untreated and 200 MPa treated oat okara. The water holding capacity did not change as a function of high-pressure treatment (3.5–3.8 mL/g) except for the treatment at 200 MPa, which was reduced (2.7 mL/g). The oil holding capacity was constant (1.2–1.3 mL/g) after all treatments. High-pressure pasteurization of 600 MPa reduced the microbial content in oat okara resulting in a shelf life of 2–4 weeks. However, more research is required to identify the microorganisms in oat okara to achieve a microbiologically safe product that can be used for food applications.

## 1. Introduction

One of the largest human impacts on the environment is the agriculture system. It stands for 26% of the total greenhouse gas emissions and uses 50% of all habitable land (ice and desert-free). Of all crops that are cultivated globally, 77% of global farming land is used for animal feed; however, it only contributes to 18% of the world’s calories [1]. The extensive land use combined with concerns about health and animal welfare motivates consumers, especially in developed countries, to change diet lifestyles toward a more plant-based diet [2]. The plant-based dairy analog market was valued at approximately USD 27.0 billion in 2023 and is expected to increase at a 10.1% rate until 2028 [3]. Soy beverages have the largest share of the market but oat beverages have been an expanding market in recent years [4] and were the second largest segment in US retail in 2018 [5].

The production of oat beverages often starts with the soaking of the oat kernels followed by a grinding step. Thereafter, the oats undergo an enzymatic treatment which is required to reduce the viscosity and increase the yield of the beverage [6]. The starch is consequently converted by enzymes into sugars which improve the texture, stability, mouthfeel, and flavor. The enzymes are thereafter inactivated by thermal treatment at 95 °C for 80 s. The inactivation is followed by a liquid–solid separation with a decanter that separates the oat beverage from its fiber-rich residue, oat okara [7]. For every processed kilogram of oats, approximately 0.45 kg of oat okara [8] is co-produced. The market leader in the oat beverage segment has a co-production of up to 84,000 tons of oat okara globally per year. The oat okara is mainly converted into energy as biomethane or by incineration or is used as animal feed or soil improvement [9].

The oat okara has a high protein content (24–32% dry basis) and a high fiber content (26–35% dry basis) which makes it nutritionally valuable [10,11]. This by-stream has the potential to be utilized in food applications but, due to its high water content of approximately 65% [10,11], it has a short shelf life.

To be able to utilize oat okara as food, it must be microbiologically stabilized to assure food safety. The most common method for the reduction in microorganisms in foods is thermal pasteurization; however, due to the high viscosity and fibrous content, heat exchangers would not be suitable. Nonetheless, other types of pasteurization methods are available.

High-pressure pasteurization (HPP) is an alternative to thermal pasteurization where a vacuum-packed material is exposed to high pressures in a pressure vessel. The pressure is applied with a pressure-transmitting fluid, typically water, and the fluid is compressed with a pump and intensifier. The volume of the packaged material can be reduced to up to 15% of its original volume during treatment but is afterward essentially restored. A typical process time is around 5 min and the pressure applied is usually between 200–600 MPa [12]. There are large investment costs for HPP but it requires less processing time and energy consumption compared to conventional thermal methods [13]. However, it is performed in a batch-wise manner which makes it difficult to apply the technique on high-speed production lines. There is still a lack of basic data on production conditions for HPP foods, such as pressure-resistant characteristics and indicator microorganisms for food safety [12]. It is therefore critical to know the optimal process conditions for a specific material to increase efficiency and safety outcomes.

HPP is commonly applied to food products such as fruit juices and guacamole where vitamins, flavors, and color are desirable attributes. Compared to thermal pasteurization, HPP reduces viable microorganisms but does not alter the sensory or nutritional qualities; the food products are therefore kept in a fresh-like state [13]. High pressure has previously been applied on soy okara (the residue from soy beverage production) to improve the functionality of the dietary fiber, to increase the soluble dietary fiber fraction [14,15], or to improve the shelf life [16]. However, to the authors’ knowledge, high pressure has not yet been applied to oat okara.

The microbiological inactivation mechanisms of HPP are not yet fully understood but it is believed that the treatment can damage cell membranes, nucleoids, ribosomes, and proteins [13]. The denaturation of proteins is caused by water that is pushed into the proteins’ core by pressure which destabilizes the hydrophobic interactions and forces proteins to unfold [17]. However, the primary and secondary protein structures are negligibly affected due to their covalent bonds which are resistant to pressure [18].

High pressure has no or little effect on molecules that are associated with food qualities such as nutritional content, flavor, and color. However, the structure of larger molecules such as polysaccharides, nucleic acids, proteins, and enzymes can be altered [19]. Physical and functional properties of an HPP-treated material can therefore change, such as the melting point, solubility, density, and viscosity [13]. It is implied that different microorganisms have different sensitivities to HPP, where the most sensitive is gram-negative bacteria and thereafter yeasts and gram-positive bacteria, with bacterial spores being the most resistant [20].

This study aimed to evaluate the effect of HPP on the microbiological stability in oat okara as well as functional properties such as endothermal transitions, viscosity, and water and oil holding capacities. If oat okara were to be efficiently pasteurized by HPP, the prolonged shelf life could open up new opportunities for oat okara as a food product or ingredient instead of being used as animal feed or being converted into energy.

## 2. Materials and Methods

### 2.1. Materials and Chemicals

Oat okara was kindly provided by The Green Dairy (Karlshamn, Sweden) and was analyzed directly (Crude oat okara, Figure 1) or stored in a freezer (−18 °C) until further analysis (Thawed and vacuum-packed, Figure 1) due to practical reasons.

Malt Extract Agar (MA) (Sigma-Aldrich, St. Louis, MO, USA), Tryptic soy agar (TSA) (Sigma-Aldrich, St. Louis, MO, USA), De Man Rogosa and Sharpe agar (MRS) (Merck, Darmstadt, Germany) and Violet Red Bile Dextrose agar (VRBD) (Merck, Darmstadt, Germany) were used for microbiological analysis. The samples were diluted in peptone water, 0.1% (Oxoid, Hampshire, UK).

Rapeseed oil (ICA Sweden AB, Solna, Sweden) was purchased at a local supermarket for the measurement of oil holding capacity.

### 2.2. Experimental Design of HPP-Process and Storage Study

Frozen oat okara samples were thawed in a water bath, vacuum-packed in three separate PA/PE bags in 200 g batches (Multivac A300/11, Sepp Haggenmüller SE and Co. KG, Wolfertschwenden, Germany), and transported in a cooling bag with ice packs (3–4 h in total) from the laboratory in Lund (Sweden) to HPP Nordic (Landskrona, Sweden). A high-pressure processor (AV-10, AVURE Technologies, Erlanger, KY, USA) was used for the HPP treatments. The vacuum-packed oat okara was pressurized at 200 MPa, 400 MPa, and 600 MPa for 3 min (a common commercial holding time [21]) in triplicate bags. The temperature and pressure intervals are presented in Table 1.

The storage study and microbiological analysis were performed on the HPP-treated oat okara which was incubated for 2 weeks and 4 weeks (4 °C); the untreated oat okara, which was incubated for 4 h, 2 weeks, and 4 weeks (4 °C), is referred to as the reference. The 4 h storage time for the reference was used to mimic the transportation back and forth to the HPP facility and represented the microbiological starting point before HPP treatment. Crude oat okara, freshly produced from the Green Dairy, was also analyzed to evaluate the effect of freezing on the microbiological load. Oat okara was packed after production in the factory and was transported at 4 °C and arrived at the lab after 18 h. The storage study design has been summarized in Figure 1.

**Figure 1 foods-12-04070-f001:**
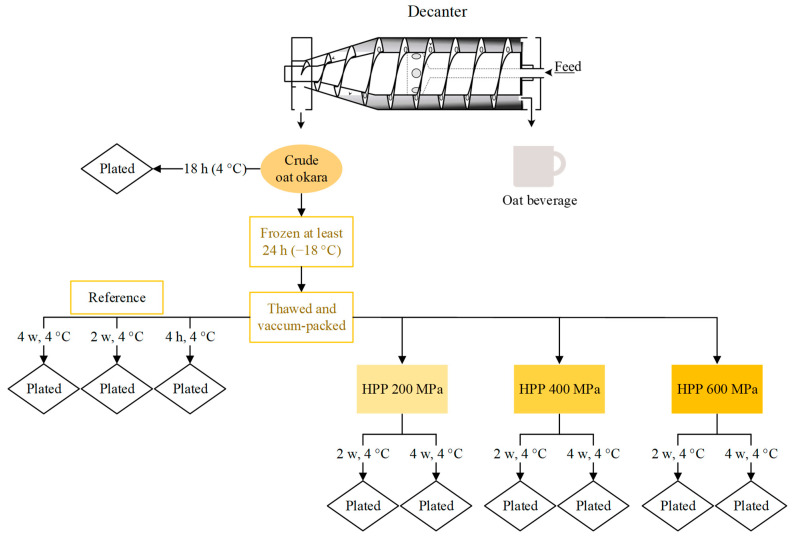
The storage study design for microbiological analysis for reference, HPP-treated, and crude oat okara (decanter image adapted with permission from Ref. [22]. 2023, Alveteg).

### 2.3. Microbiological Content

Each sample was weighed (10 g) and added in a sterile bag together with 90 mL of 0.1% peptone water and was homogenized with a stomacher (Seward BA6021 stomacher, Bury St. Edmunds, UK) for 1 min. Appropriate serial dilutions with 0.1% peptone water were applied. In each petri dish, 100 µL of a dilution was added and spread over the surface with sterile glass beads. Each dilution level was plated in duplicate. The TSA plates were incubated aerobically at 30 °C for 72 h ± 6 h (total aerobic count), MA incubated aerobically at 25 °C for 5–7 days (yeast and mold), MRS was incubated anaerobically at 37 °C for 72 h ± 6 h (lactic acid bacteria), and VRBD was incubated aerobically at 37 °C for 24 h ± 1 h (Enterobacteriaceae). The lower detection limit for the analysis was 2.30 log CFU/g and 20–300 colonies on one petri dish were considered to be acceptable.

### 2.4. Proximate Composition

The oat okara reference was analyzed at Eurofins Food and Feed Testing, Sweden, Lidköping, for its proximate composition. Analysis methods were based on the Nordic-Baltic Committee on Food Analysis (NMKL). The protein was analyzed according to the Kjeldahl method (Nx6.25), the fat according to NMKL 160 mod., the carbohydrates were calculated by difference (EU) nr 1169/2011, the fiber according to AOAC 991.43 mod., the ash according to NMKL 173, and the water according to NMKL 23. To investigate any changes in the soluble and insoluble fiber ratio due to the HPP treatments, both reference and HPP-treated samples were analyzed for soluble and insoluble fiber content (AOAC 991.43 mod.) at Eurofins. The samples were analyzed in one replicate and the standard deviation in the measurements was based on the measurement error.

Reference and HPP-treated oat okara were analyzed on-site for water content according to AACC 44–15A. The sample (3–5 g) was weighed before and after drying in an oven at 103 °C until constant weight in a metal container. The analysis was performed in triplicate.

### 2.5. Water Activity

Reference and HPP-treated oat okara were analyzed with a water activity meter (AquaLab Ver 3TE, Decagon Devises, Pullman, WA, USA) at 20 °C to analyze the water activity. The instrument was calibrated with standard salt solutions (13.41 M LiCl (0.250 a_w_), 8.57 M LiCl (0.500 a_w_), and 6 M NaCl (0.760 a_w_)). The analysis was performed in triplicate.

### 2.6. Differential Scanning Calorimetry

The thermal properties were investigated with differential scanning calorimetry (DSC) on reference and HPP-treated oat okara (Seiko 6200 DSC, Seiko Instruments Inc. Shizuoka, Japan). Oat okara (4–6 mg) was weighed into aluminum-coated pans. Milli-Q water was added to the oat okara (1:1 *w*/*w*) to reach a water content of approximately 75%. The pans were sealed and thereafter run in the DSC according to the settings presented in Table 2. An empty pan was used as a reference. A holding time of 1 min was applied to achieve a stable baseline for all samples. After the DSC analysis, the dry matter content was analyzed by puncturing the pans and thereafter letting the pans dry in an oven at 105 °C until constant weight. Analysis was performed in at least triplicate.

### 2.7. Viscosity

A high-temperature Rapid Visco Analyzer (RVA 4800, Perkin Elmer, Waltham, MA, USA) was used to analyze the viscosity of reference and HPP-treated oat okara. Oat okara was run at a moisture content of 85% (10 g sample and 18.67 g deionized water) with the test profile presented in Table 3. Analysis was performed in triplicate.

### 2.8. Water and Oil Holding Capacities

Reference and HPP-treated oat okara were analyzed for water and oil holding capacity (WHC and OHC) according to the method described by Ho et al. [23] with a few modifications. For WHC, 10 g of oat okara was weighed into Falcon tubes (50 mL) together with approximately 40 mL of deionized water. For OHC, 2 g of oat okara was added to 20 mL of rapeseed oil (ICA Sweden AB, Solna, Sweden). The blends were vortexed for 2 min and thereafter incubated at room temperature for 30 min. After incubation, the samples were centrifuged at 4900× *g* for 20 min (Beckman Coulter, Avanti^®^J-15R Centrifuge, Brea, CA, USA). The supernatant was decanted and the Falcon tube and pellet were reweighed. The analysis was performed in triplicate. The WHC was calculated according to Equation (1) and OHC was calculated according to Equation (2).
(1)$$WHC=\frac{m\left(water\;content\;in\;sample\right)+m\left(test\;tube+pellet\right)-m\left({test\;tube+sample}_{dry\;solid}\right)}{m({sample}_{dry\;solid})}\frac{mL}{g_{dry\;solid}}$$
(2)$$OHC=\frac{m\left(test\;tube+pellet\right)-m(test\;tube+sample)}{m(sample)}\frac{mL}{g}$$

### 2.9. Scanning Electron Microscopy

Frozen samples of reference and HPP-treated oat okara were freeze-dried (Labconco, MO, USA) and milled. The material was gently glued onto scanning electron microscopy (SEM) stubs and sputter-coated with gold (Cesington 108 auto, 120 s, 20 mA). The samples were imaged using a scanning electron microscope (SEM; Hitachi, SU3500, Tokyo, Japan) at 5 kV.

### 2.10. Statistical Analysis

The statistical analysis was performed in Minitab. A normality test was executed on each data set. Significant differences were detected with the Tukey test for normally distributed data and the Games-Howell test for non-parametric data. *p*-values < 0.05 were considered significantly different.

## 3. Results and Discussion

### 3.1. Microbiological Content

The microbiological content was analyzed for reference and HPP-treated oat okara after two weeks and four weeks to evaluate whether the HPP treatment had any significant effect on the shelf life of oat okara. Enterobacteriaceae was below the detection limit (<log 2.3 CFU/g) for the crude sample and reference after two weeks and was therefore not further investigated in the storage study for the HPP samples. The reference had a significantly lower (*p* < 0.05) microbial content compared to the crude oat okara on all agar types prior to HPP treatment (Figure 2A–C) which indicates that the freezing treatment influenced the microbial load in the samples.

The content of the total aerobic count was significantly reduced (*p* < 0.05) by the treatment of 600 MPa after two weeks (log 4.0 CFU/g) and four weeks (log 5.2 CFU/g, Figure 2A) compared to the reference. The treatments of 200 MPa and 400 MPa significantly enhanced (*p* < 0.05) the content of the total aerobic count after two weeks compared to the reference. After four weeks, the growth seemed to reach a plateau for the 200 MPa treatment (log 7.2 CFU/g), closely followed by the 400 MPa treatment (log 6.6 CFU/g) and the reference (log 6.6 CFU/g). After four weeks, oat okara treated at 600 MPa had the lowest microbial content.

For yeast and mold, the microbial content was below the detection limit after two weeks for all treatments (log 2.3–2.8 CFU/g Figure 2B). After four weeks, the CFU for yeast and mold increased for the reference, 200 MPa, and 400 MPa treatment (log 3.7–4.6 CFU/g); however, the microbial content was still below the detection limit for the 600 MPa treatment (log 2.3 CFU/g).

Possible explanations for the increased microbial content at 200 MPa are that oligomeric proteins can dissociate after exposure to high pressures but can often recover and reassociate after treatment [24]. The cell wall and membrane disruption by HPP can also increase the bioavailability of nutrients which can increase substrate accessibility for surviving microorganisms [13]. Spores are difficult to inactivate with HPP. At 600 MPa, Shigehisa et al. [20] reported that the growth of *Bacillus cereus* spores could only be reduced by 1 log CFU/g. However, HPP can trigger spore germination, making the bacteria vulnerable. In this way, HPP can become lethal to spore-forming bacteria [25]. The germination effect by HPP could explain the significantly higher growth of total aerobic count for 200 MPa and 400 MPa treatment compared to the reference. Hence, the pressures at 200 MPa and 400 MPa may not have been high enough to both germinate and inactivate spores. This was, however, probably possible for the HPP treatment of 600 MPa as a reduction in total aerobic count was observed for this pressure.

The growth of lactic acid bacteria followed approximately the same trend as for yeast and mold; however, the treatment of 400 MPa had the same effect as the 600 MPa treatment with microbial content below the detection limit (Figure 2C). A growth reduction in lactic acid bacteria at 400 MPa and 600 MPa treatments has been observed before in a previous study [26]. Even though lactic acid bacteria are gram-positive, they did not have a stronger resistance to HPP in comparison to yeast and mold in this study. However, the sensitivity also depends on the strain of bacteria and the growth cycle the microorganisms are in when they are exposed to HPP treatment [13].

The shelf life of oat okara treated with 600 MPa was 2–4 weeks when refrigerated at 4 °C (log 4.0–5.2 CFU/g). However, after four weeks of storage, the microbial content was within the upper acceptance range (log 5–6 CFU/g [27,28]). Oat okara with no treatment could also be considered safe to eat after two weeks of storage (log 4.9 CFU/g) but not safe to eat after four weeks as it was above the upper acceptance range (log 6.6 CFU/g). A shelf life of 1–2 weeks would therefore only require freezing and vacuum-packing but, if a longer shelf life would be required (2–4 weeks), an HPP treatment of a minimum of 600 MPa could be considered. HPP treatment of 200 MPa and 400 MPa would not be appropriate as these treatments instead increased the microbial growth. In future research, it would be interesting to analyze the bacteria species more specifically to be able to know which type of bacteria is contributing most to bacterial growth. If spore-forming bacteria would be present to a large extent, a combination of 600 MPa and heat, also known as pressure-assisted thermal sterilization [13], would be recommended for investigation.

### 3.2. Proximate Composition and Water Activity

Microorganisms require sources of energy in the form of carbohydrates, proteins, and fats. Generally, plant products contain limiting amounts of protein [29]; however, the proximate composition of the oat okara in this study showed a high protein content of 52% (Table 4) which can be favorable for microbial growth. In comparison to other studies on oat okara, this was a high protein content with comparable studies reporting protein content of around 24–32% [10,11]. The fiber content in the present study (16%) was low compared to other studies on oat okara which have been reporting fiber contents around 26–35% [10,11]. The difference in proximate composition can be due to differences in oat cultivars and production methods [30].

Microbial growth is highly dependent on the water content and water activity of a material. The average water content for all samples was 56.9 ± 0.2%. The water activity was 0.988 ± 0.002 for the reference and the HPP treated samples were not significantly different (*p* > 0.05). The water activity was therefore not affected by the HPP treatment.

In the reference sample, the total dietary fiber consisted only of insoluble fiber. This is a conflicting result compared to a study by Aiello et al. [10] where untreated oat okara had a soluble fiber fraction of 12% of the total dietary fiber. However, with increasing pressure, the soluble fiber fraction increased to 4% (200 MPa), 6% (400 MPa), and 9% (600 MPa) of the total dietary fiber (Table 5). A redistribution from insoluble to soluble fiber has been observed in an earlier study where high hydrostatic treatment (at 30 °C and 60 °C) increased the soluble fiber content in soy okara [14]. This redistribution can in turn affect physical properties such as water and oil holding capacities. Soluble fiber also have positive physiological effects such as a reduction in glycemic response and plasma cholesterol [31].

### 3.3. Thermal Properties

The thermal properties were investigated to evaluate the effect of the HPP treatments on the native proteins in oat okara. The water content of the analyzed samples was between 75 and 76%.

The 12S globulin in oat, avenalin, has a protein denaturation temperature of around 114–116 °C [32,33] which is represented in the thermogram for the reference with an enthalpy of 7.52 mJ/mg at 111.7 °C (Figure 3). Avenalin is an oligomeric protein [32] and at 200 MPa, its subunits may have dissociated and after treatment been able to reassociate. However, hysteresis of conformational changes can occur when this happens [24]. This could explain the broader peak and lower enthalpy (4.58 mJ/mg) that was observed for 200 MPa (Figure 3) with a higher diversity of conformational structures of the protein present. At 400 MPa and 600 MPa, the enthalpy decreased even more to 1.23 mJ/mg and 0.13 mJ/mg, respectively, which means that an increase in irreversible denaturation of proteins occurred with increasing pressure. At 600 MPa, another endothermic peak appeared at a higher temperature (133.1 °C) which possibly represents protein aggregates formed by the protein–protein interaction of unfolded proteins [34].

HPP treatment was conducted to induce conformational changes in the proteins as this is one of the main microbial inactivation mechanisms. The effect was proportional to pressure where the highest pressure (600 MPa) induced the largest protein denaturation which corresponds to the reduction in microbial content in Section 3.1. Protein denaturation can affect functional properties and it would not be recommended to use HPP-treated oat okara for more complex food products such as extruded meat analogs as a main protein ingredient as it is suggested that proteins should be in their native state to be able to unfold and aggregate in the extrusion process [35].

### 3.4. Viscosity

The HPP treatment induced protein denaturation in the oat okara (Figure 3) which changed the protein structure. HPP can also stabilize denatured proteins and induce the formation of protein aggregates. This affects functional properties such as gelation and solubility [13]. The pasting and viscosity properties of oat okara were therefore investigated to evaluate the physical effects of the HPP treatments.

For the reference and HPP-treated oat okara at 200 MPa, a viscosity peak of around 120 °C was observed (Figure 4). The peak corresponds well to the protein avenalin which was identified in the DSC thermograms (Figure 3). For HPP treatments of 400 MPa and 600 MPa, the viscosity peak disappeared which confirms altered protein conformation and protein denaturation. The noise for 600 MPa at the start of the measurement could be due to the physical resistance of highly compressed oat okara containing larger chunks of material.

In earlier RVA measurements (up to 95 °C) performed on oat flour, a gelatinization peak at 95 °C (approximately 3000 cP) has been observed [33]. That gelatinization peak was not detected in this study, indicating that all starch in oat okara has already been gelatinized in the oat beverage production process.

### 3.5. Water and Oil Holding Capacities

The HPP treatment at 200 MPa gave a significantly lower WHC (*p* > 0.05) compared to reference as well as HPP-treated oat okara at 400 MPa and 600 MPa (Figure 5). The WHC of solid foods is dependent on the 3D network of biopolymers, usually proteins or polysaccharides [7]. As conformational changes of the proteins after the 200 MPa treatment (Figure 3) probably occurred, it could explain the loss of WHC as the altered proteins might have affected the three-dimensional network. Even though an increase in protein denaturation followed the treatments of 400 MPa and 600 MPa, the WHC recovered and was as high as the reference. The explanation for this could instead be dependent on the polysaccharides. As there was an incremental increase in soluble fiber in the HPP-treated samples (Table 5), this fiber rearrangement might be an explanation to the increase in WHC from 200 MPa to 400 MPa and 600 MPa, which could have compensated for the possible loss of 3D network of proteins due to protein denaturation. The same phenomenon has been observed in an earlier study where soy okara had been subjected to high hydrostatic pressure (200 MPa and 400 MPa) and an increase in soluble fiber as well as water retention capacity was observed [14].

The HPP treatments did not affect the OHC for oat okara (*p* > 0.05). The OHC was between 1.2–1.3 mL oil/g (Figure 5), which is slightly higher than wheat flour with an OHC of around 0.88 mL oil/g [23]. As was observed in Section 3.3, increased pressure increased protein denaturation, concluding that the extent of denatured proteins in the oat okara was not related to the OHC. It was therefore possible to reduce the microbial load in oat okara without affecting the OHC with a 600 MPa HPP treatment.

### 3.6. Scanning Electron Micrographs

The gelatinization of starch can be induced by high pressures and, at pressures above 500 MPa, starch will start to gelatinize [13]. However, as was discussed in Section 3.4, most starch in oat okara has already been gelatinized in the oat beverage production process which can also be visualized by the coating of gelatinized starch in the micrographs for all samples (Figure 6b,d,f,h).

A mixture of sheets with gelatinized starch and cell wall structures together with smaller particles were observed for all treatments (Figure 6). However, at the highest pressure of 600 MPa, the cell wall structure was ruptured, generating a flaky structure (Figure 6g).

## 4. Conclusions

The HPP treatment of oat okara at 600 MPa successfully reduced the microbiological content of total aerobic count as well as yeast and mold after four weeks of storage (vacuum-packed, 4 °C). The content of lactic acid bacteria was not excessively high in the reference to start with (3.2 log CFU/g) but was kept at the detection level (2.3 log CFU/g) with the 600 MPa treatment. The aerobic plate count for the 600 MPa treatment (log 5.2 CFU/g) was within the upper acceptance range (log 5–6 CFU/g) after four weeks, which could imply an issue of spore-forming bacteria. If this would be the case, a possible solution would be to combine high pressure with heat, also known as pressure-assisted thermal sterilization.

Increasing HPP pressures increased denaturation of the 12S globulin protein avenalin which, in turn, decreased viscosity properties of the oat okara. Increased pressure did, however, also increase the soluble fiber content which has positive physiological effects such as reduction in glycemic response and plasma cholesterol [31]. The WHC was affected by the 200 MPa treatment where conformational changes and the denaturation of proteins might have affected the 3D network. The WHC was, however, similar to the reference level after the 400 MPa and 600 MPa treatments, probably by compensation of an increase in soluble fiber and fiber rearrangement. The OHC was not affected by the high-pressure treatments. Even though HPP-treated oat okara has more denatured proteins and lower apparent viscosity, it still has a high WHC and OHC and could therefore be used as a food ingredient in bread, pastries, and other food products to increase fiber and protein content.

## Figures and Tables

**Figure 2 foods-12-04070-f002:**
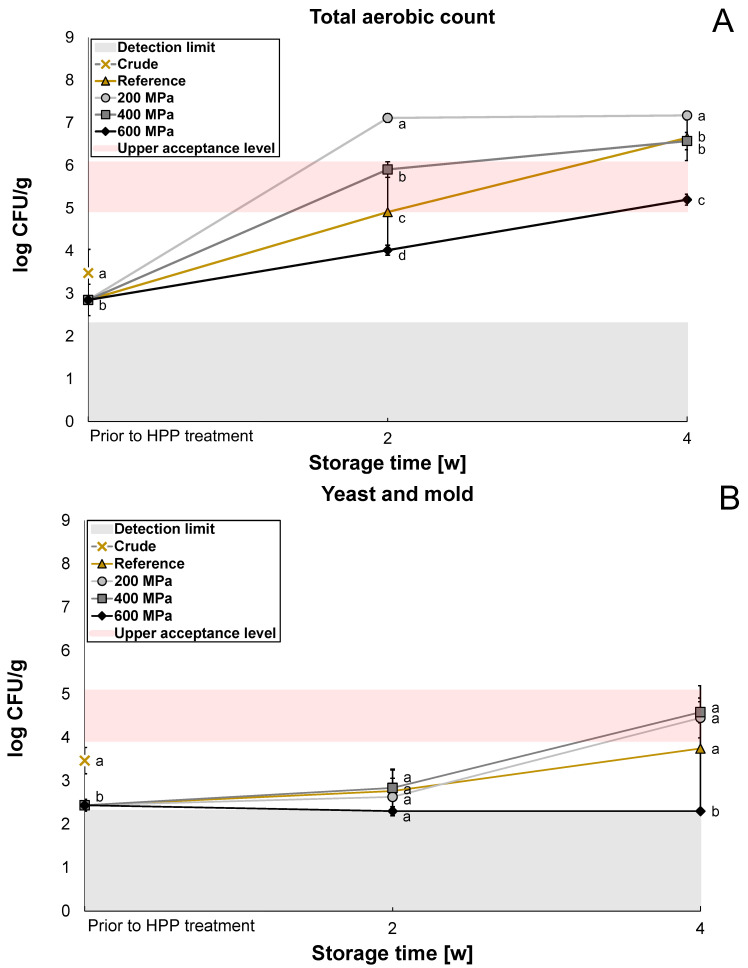
The microbiological content of crude oat okara after 18 h at 4 °C (cross), reference after 4 h, 2 weeks, and 4 weeks at 4 °C (triangle), and HPP-treated oat okara at 200 MPa (circle), 400 MPa (square), and 600 MPa (diamond) after 2 weeks and 4 weeks at 4 °C. Presented as the total aerobic count (**A**), yeast and mold (**B**), and lactic acid bacteria (**C**). Statistical comparisons of the microbial content were made at each storage time for each agar type. Data with different letters are significantly different, *p* < 0.05, *n* = 6.

**Figure 3 foods-12-04070-f003:**
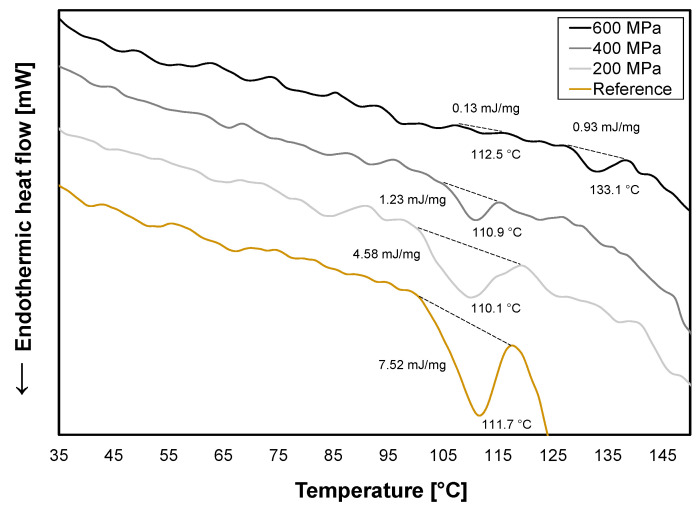
DSC curves for reference (yellow) and HPP-treated oat okara at 200 MPa (light gray), 400 MPa (gray), and 600 MPa (black).

**Figure 4 foods-12-04070-f004:**
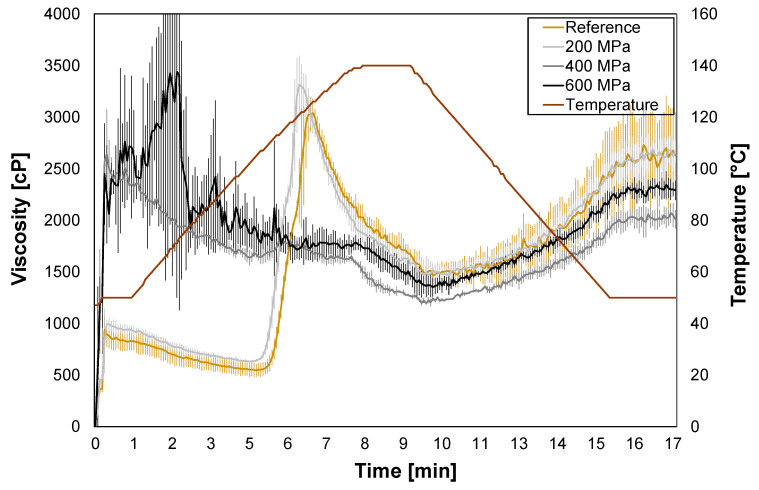
RVA graphs presenting average viscosity for reference (yellow) and HPP-treated oat okara at 200 MPa (light gray), 400 MPa (gray), and 600 MPa (black).

**Figure 5 foods-12-04070-f005:**
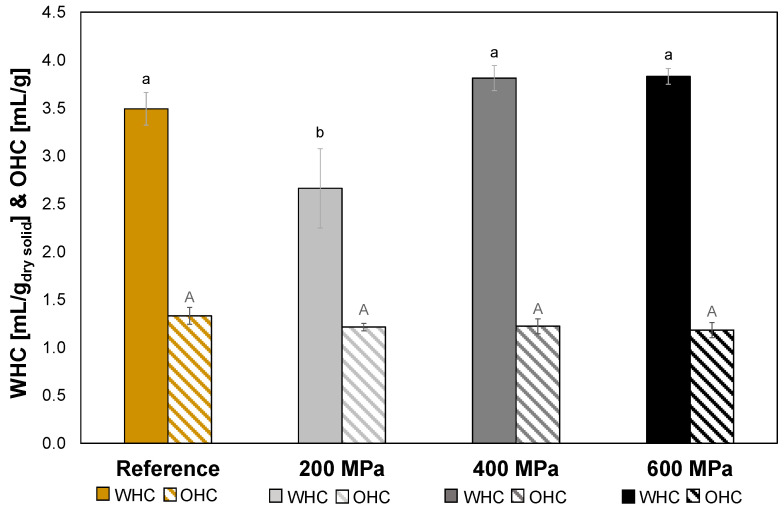
WHC (solid) and OHC (striped) results for reference (yellow) and HPP-treated oat okara at 200 MPa (light gray), 400 MPa (gray), and 600 MPa (black). Data with different letters are significantly different, *p* < 0.05, *n* = 3 (lower case letters for WHC, and upper case letters for OHC).

**Figure 6 foods-12-04070-f006:**
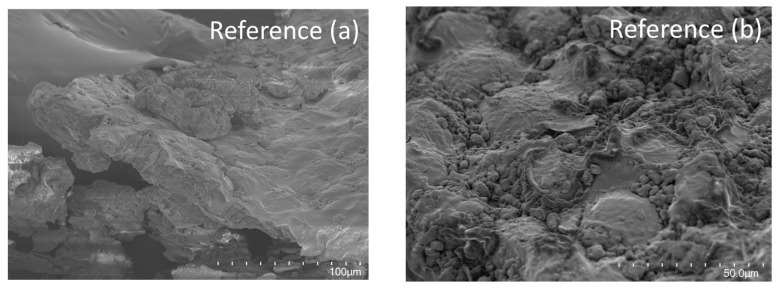
SEM micrographs of reference (**a**,**b**) and HPP-treated oat okara at 200 MPa (**c**,**d**), 400 MPa (**e**,**f**), and 600 MPa (**g**,**h**). Micrographs were taken at magnifications of 500× (first column, (**a**,**c**,**e**,**g**)), and 1000× (second column, (**b**,**d**,**f**,**h**)).

**Table 1 foods-12-04070-t001:** The HPP treatment protocol.

Pressure (MPa)	Temperature Interval (°C)	Pressure Interval (MPa)
200	16.2–20.7	199.8–206.3
400	18.7–20.8	400.6–404.7
600	21.5–22.6	600.7–605.2

**Table 2 foods-12-04070-t002:** DSC settings.

	Start (°C)	Limit (°C)	Rate (°C/min)	Hold (min)	Sampling (s)
	25	25	10	1	0.2
End step	25	200	10	0	0.2

**Table 3 foods-12-04070-t003:** The RVA test profile.

Time		
00:00	Temp	50 °C
00:00	Speed	960 rpm
00:10	Speed	160 rpm
01:00	Temp	50 °C
06:50	Temp	140 °C
09:20	Temp	140 °C
15:10	Temp	50 °C
17:10	End	

**Table 4 foods-12-04070-t004:** Proximate composition of oat okara on dry basis.

Protein (%)	Fat (%)	Carbohydrate (%)	Total Dietary Fiber (%)	Ash (%)	Water (%)
52.1 ± 5.2	14.1 ± 1.4	12.3 *	15.8 ± 2.4	5.8 ± 0.6	54.3 ± 5.4

* Calculated by difference, *n* = 1, the standard deviation was based on the measuring uncertainty of ±10%, except for total dietary fiber with ±15% uncertainty.

**Table 5 foods-12-04070-t005:** Fiber content, as well as the insoluble and soluble fiber content in oat okara for all treatments on dry basis.

	Reference	200 MPa	400 MPa	600 MPa
Total dietary fiber (%)	15.8 ± 2.36	18.1 ± 2.72	17.4 ± 2.62	18.5 ± 2.78
Insoluble fiber (%)	16.0 ± 2.40	17.4 ± 2.61	16.3 ± 2.45	16.8 ± 2.52
Soluble fiber (%)	n.d.	0.7 ± 0.11	1.1 ± 0.17	1.7 ± 0.26

n.d. = not detectable, *n* = 1, the standard deviation was based on the measurement uncertainty of ±15%.

## Data Availability

The datasets generated during the current study are publicly available from the corresponding author upon request.

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
