# Peer review of "High-Pressure Pasteurization of Oat Okara"

_foods, 2023, doi:10.3390/foods12224070_

Round 1
Reviewer 1 Report
Comments and Suggestions for Authors
Minor corrections
DSC result can be further analyzed to obtain other properties such as change in enthalpy (
) and change in specific heat capacity (
). Moreover, the rheological properties such as shear stress, shear rate and shear deformation can also be included in the rheology aspect of the study. I suggest the authors assess the effect of HPP treatment on the textural attributes of the okara. Lastly, the authors should compare their findings with already published results.
Line 11: the pressure unit should be with the last figure only.
Author Response
Dear Reviewer 1
Thank you for the valuable comments. We have done our best to make the suggested changes and hope you find them satisfactory in the revised manuscript. Please find our responses to the comments below. In the revised manuscript new and significantly modified sections are indicated with red font.
Comments and Suggestions for Authors
Comment #1 DSC result can be further analyzed to obtain other properties such as change in enthalpy () and change in specific heat capacity (). Moreover, the rheological properties such as shear stress, shear rate and shear deformation can also be included in the rheology aspect of the study. I suggest the authors assess the effect of HPP treatment on the textural attributes of the okara. Lastly, the authors should compare their findings with already published results.
Response: We have added more information about the enthalpy changes in Section 3.3 (p. 10 lines 337 and 339-340), however, we have not chosen to include the specific heat capacity since it is not relevant for this study. The RVA only provides information on apparent viscosity, therefore we do not have the rheological properties of shear stress, shear rate, and shear deformation. Furthermore, the focus of the study is to investigate the effect of HPP on oat okara to be used as an ingredient in the food industry, for example in baking or meat analog applications. Therefore, it is in our view more interesting to measure apparent viscosity in the oat okara under the influence of heat and shear, rather than textural attributes on oat okara as such.
Comment #2: Line 11: the pressure unit should be with the last figure only.
Response: Thank you for this comment, we have followed the SI Unit rules and style conventions (https://physics.nist.gov/cuu/Units/checklist.html) which recommend putting the unit behind each figure. In cases where pressure intervals have been reported the unit is only after the last figure. We have used this principle throughout the manuscript.

Reviewer 2 Report
Comments and Suggestions for Authors
The manuscript foods-2684667 shows the effect of HPP on the microbiological stability in oat okara, as well as functional properties such as endothermal transitions, viscosity, and water and oil holding capacities.
I consider that the paper has been well written, however, it does not show conclusive results, since it does not demonstrate the increase in shelf life with the HPP process. Furthermore, it gives confusing results, since in some cases the HPP process even seems to worsen the shelf life of the raw material. In general, I have the impression, based on the results, that it would be better not to apply the HPP treatment, which is quite expensive. On the other hand, situations that were not studied in this work, such as the inactivation of Bacillus cereus, are discussed. I also invite the correct and homogeneous expression of the units of measurement.
Comments:
L16: CFU/g. Correct throughout the text.
L18: Specify pressure.
L21: L (liter) should be capitalized throughout the text.
L24: Specify how long.
Introduction: Comment on similar studies in the Introduction to indicate the novelty of the work.
L112: Indicate time ranges to be more precise.
L136: 72 h ± 6 h or (72 ± 6) h. Correct throughout the text for all cases of units, as the problem is repeated continuously.
L141: References to proximal methods do not appear.
L210-211: I think it refers to Figure 2.
L213-219: A proper explanation needs to be given for this behavior. How do you explain the increased total aerobic counts (Figure 2a) when applying 200 MPa and 400 MPa compared to the reference? Was there contamination of the samples treated with 200 MPa and 400 MPa? This is a major problem.
L234: Bacillus cereus
L238: B. cereus
L234-252: Bacillus cereus was not accounted for in this study, so you should remove any discussion in reference to this microorganism. This is another major problem.
L257: It would be safe after 2 and 3 weeks according to Figure 2, so the whole paragraph should be modified.
L277-279: Are they comparing with the same material? Otherwise there is no point in comparison. The studies are not included so it is impossible to verify.
L285-300: It does not make any analysis of the relationship of the proximal content with the microbiological results. The results have to be discussed in an integrative way.
L278: Which studies?
Table 4: No significant figures are reported.
L350: There was also no difference with the reference.
L356: Are there any references to support this?
L362-369: As indicated by the authors in L369-372, this is not demonstrated in this work.
Author Response
Dear Reviewer 2
Thank you for the valuable comments. We have done our best to make the suggested changes and hope you find them satisfactory in the revised manuscript. Please find our responses to the comments below. In the revised manuscript new and significantly modified sections are indicated with red font.
Comments and Suggestions for Authors
The manuscript foods-2684667 shows the effect of HPP on the microbiological stability in oat okara, as well as functional properties such as endothermal transitions, viscosity, and water and oil holding capacities.
I consider that the paper has been well written, however, it does not show conclusive results, since it does not demonstrate the increase in shelf life with the HPP process. Furthermore, it gives confusing results, since in some cases the HPP process even seems to worsen the shelf life of the raw material. In general, I have the impression, based on the results, that it would be better not to apply the HPP treatment, which is quite expensive. On the other hand, situations that were not studied in this work, such as the inactivation of Bacillus cereus, are discussed. I also invite the correct and homogeneous expression of the units of measurement.
Comment #1: L16: CFU/g. Correct throughout the text.
Response: Thank you for this comment, the unit has been corrected throughout the manuscript.
Comment #2: L18: Specify pressure.
Response: Thank you, the pressure has been specified (p. 1, line 20).
Comment #3: L21: L (liter) should be capitalized throughout the text.
Response: Liter has been capitalized throughout the manuscript.
Comment #4: L24: Specify how long.
Response: Thank you, we have now specified the length of shelf life after 600 MPa treatment (p. 1, lines 25-26).
Comment #5: Introduction: Comment on similar studies in the Introduction to indicate the novelty of the work.
Response: Thank you, we have now added more references in the introduction mentioning high-pressure treatments on the soy beverage residue soy okara (p. 2, lines 77-81). To our knowledge, high pressure has not been applied to oat okara in any earlier study – indicating the novelty of this study.
Comment #6: L112: Indicate time ranges to be more precise.
Response: We have indicated a time range (p. 3, line 117).
Comment #7: L136: 72 h ± 6 h or (72 ± 6) h. Correct throughout the text for all cases of units, as the problem is repeated continuously.
Response: Thank you, this has been corrected.
Comment #8: L141: References to proximal methods do not appear.
Response: Due to corona restrictions at our institution, samples were sent to the accredited laboratory Eurofins which uses analysis methods based on the Nordic-Baltic Committee on Food Analysis (NMKL) which can be found on their website if purchased. We have added the information about NMKL to clarify (p. 4, lines 152-156).
Comment #9: L210-211: I think it refers to Figure 2
Response: Thank you for this observation, it is now correctly referred to Figure 2.
Comment #10: L213-219: A proper explanation needs to be given for this behavior. How do you explain the increased total aerobic counts (Figure 2a) when applying 200 MPa and 400 MPa compared to the reference? Was there contamination of the samples treated with 200 MPa and 400 MPa? This is a major problem.
Response: Our theory about this behavior is that pressures at 200 MPa and 400 MPa might have had a germination effect on spores in the oat okara, enhancing microbial growth. At 200 MPa, oligomeric proteins can dissociate but also recover and reassociate, therefore not affecting the microorganisms too much (p. 6-7, lines 239-251).
Comment #11 & 12: L234: Bacillus cereus. L238: B. cereus
Response: We have removed discussions about Bacillus cereus. We have only mentioned Bacillus cereus once in the revised manuscript referring to another study and written it in a correct manner “Bacillus cereus” (p. 6, line 244).
Comment #13: L234-252: Bacillus cereus was not accounted for in this study, so you should remove any discussion in reference to this microorganism. This is another major problem.
Response: We have removed discussions about the probable Bacillus cereus problem and are instead focusing on spores in general (Section 3.1).
Comment #14: L257: It would be safe after 2 and 3 weeks according to Figure 2, so the whole paragraph should be modified.
Response: We have now modified the paragraph and reported a longer shelf life for the 600 MPa treatment in comparison to the reference to increase clarity (p. 7, lines 269-273).
Comment #15: L277-279: Are they comparing with the same material? Otherwise there is no point in comparison. The studies are not included so it is impossible to verify.
Response: The proximate composition has been compared to studies on oat okara (p. 9, lines 295-299).
Comment #16: L285-300: It does not make any analysis of the relationship of the proximal content with the microbiological results. The results have to be discussed in an integrative way.
Response: Thank you, we have now added and discussed the importance of protein content for microbial growth in section 3.2 as the oat okara has a high protein content (p. 9, lines 292-295).
Comment #17: L278: Which studies?
Response: References 10 and 11. It has been clarified in the manuscript (p. 9, lines 295-299).
Comment #18: Table 4: No significant figures are reported.
Response: Significant figures have been adjusted (p. 9, Table 4).
Comment #19: L350: There was also no difference with the reference.
Response: This has been clarified (p. 12, lines 375-376).
Comment #20 & 21: L356: Are there any references to support this? L362-369: As indicated by the authors in L369-372, this is not demonstrated in this work.
Response: The paragraph has been rewritten to increase clarity, and a reference for the connection between proteins/polysaccharides and WHC has been added (p. 12, lines 376-389). The cocnclusion about WHC has also been rewritten to match the discussion better (p. 15, lines 429-433).

Reviewer 3 Report
Comments and Suggestions for Authors
The paper reports the use of high pressure at 200, 400 and 600 MPa to limit microbial growth in oat okara to make it suitable for use in human food. Various physical properties of the treated okara were compared with an untreated (reference) sample. The idea of using high pressure for this application is interesting although I’m not sure about the practicality of its use commercially.
The microbiological results are curious. The treated samples and the reference all had the same total count at zero time (Fig 1A). This is not possible. It appears that the points labelled reference, 200, 400 and 600 MPa (at zero time) may not have been treated for this result. If so, this is unacceptable for zero-time data.
In lines 210-212, it should be indicated that this refers to the zero-time results
The counts for 200 and 400 MPa treatments at 2 weeks were higher than that of the untreated reference. An explanation for this result needs to be given. It is noted that microbiological data for these treatments are not mentioned in the Abstract.
The increase in total count after treatment at 600 MPa appears to be attributed to activation of spores and growth of the generated vegetative cells. B. cereus (note: cereus does not have an upper-case C) is implicated but without any proof; many other Bacillus and Bacillus-like species could be present in okara. There needs to be proof of the role of spores in the treated samples. The possible use of the pressure-treated okara in human food seems to depend on this.
The authors need to reconsider the use of the word “growth” when discussing the microbiological results. In many cases, “growth” should be substituted with “content” or “count”. For example, in lines 14 and 15, use of the word “growth” is incorrect. Also, the heading of section 2.3 is “Microbiological growth”; this should be changed to “Microbiological content” as given in line 204. See also line 387. There are several other places where the use of the word “growth” is inappropriate.
In line 77, what is the meaning of “alternate” cell membrane.
Table 5. I question the need for this table. All treatments were in closed systems so the results should be the same for all samples.
Lines 393-395 in the Conclusions. This belongs in the discussion not in the Conclusions
Comments on the Quality of English Language
Mostly OK
Author Response
Dear Reviewer 3
Thank you for the valuable comments. We have done our best to make the suggested changes and hope you find them satisfactory in the revised manuscript. Please find our responses to the comments below. In the revised manuscript new and significantly modified sections are indicated with red font.
Comments and Suggestions for Authors
The paper reports the use of high pressure at 200, 400 and 600 MPa to limit microbial growth in oat okara to make it suitable for use in human food. Various physical properties of the treated okara were compared with an untreated (reference) sample. The idea of using high pressure for this application is interesting although I’m not sure about the practicality of its use commercially.
Comment #1: The microbiological results are curious. The treated samples and the reference all had the same total count at zero time (Fig 1A). This is not possible. It appears that the points labelled reference, 200, 400 and 600 MPa (at zero time) may not have been treated for this result. If so, this is unacceptable for zero-time data.
Response: Thank you for this comment, we understand that our description was unclear. We investigated the microbial content without any HPP treatment in refrigerated storage for 4 h (to mimic transport back and forth to HPP Nordic) as a “time zero”. However, we did not investigate the microbial content directly after HPP treatments, we only investigated them after 2 weeks and 4 weeks time. We have adjusted Figure 2 and its figure legend to clarify this.
Comment #2: In lines 210-212, it should be indicated that this refers to the zero-time results.
Response: Thank you, in the comparison between reference and crude sample we have now also stated that this was prior to HPP treatment (p. 6, lines 224-225).
Comment #3: The counts for 200 and 400 MPa treatments at 2 weeks were higher than that of the untreated reference. An explanation for this result needs to be given. It is noted that microbiological data for these treatments are not mentioned in the Abstract.
Response: Our theory about this behavior is that pressures at 200 MPa and 400 MPa might have had a germination effect on spores in the oat okara, enhancing microbial growth. At 200 MPa, oligomeric proteins can dissociate but also recover and reassociate, therefore not affecting the microorganisms too much (p. 6-7, lines 239-251). We have added results of the increase of total aerobic count of 200 MPa and 400 MPa treatments in the abstract (p. 1, lines 16-18).
Comment #4: The increase in total count after treatment at 600 MPa appears to be attributed to activation of spores and growth of the generated vegetative cells. B. cereus (note: cereus does not have an upper-case C) is implicated but without any proof; many other Bacillus and Bacillus-like species could be present in okara. There needs to be proof of the role of spores in the treated samples. The possible use of the pressure-treated okara in human food seems to depend on this.
Response: Thank you for your comment, we agree and have removed discussions about the probable Bacillus cereus problem and are instead focusing on spores in general (Section 3.1).
Comment #5: The authors need to reconsider the use of the word “growth” when discussing the microbiological results. In many cases, “growth” should be substituted with “content” or “count”. For example, in lines 14 and 15, use of the word “growth” is incorrect. Also, the heading of section 2.3 is “Microbiological growth”; this should be changed to “Microbiological content” as given in line 204. See also line 387. There are several other places where the use of the word “growth” is inappropriate.
Response: Thank you for this observation, “growth” has been substituted with “content” in the appropriate places in the abstract, in the heading of Section 2.3 (p. 4), and in the conclusions as mentioned. We have reviewed the rest of the manuscript and corrected this.
Comment #6: In line 77, what is the meaning of “alternate” cell membrane.
Response: Thank you, we agree that “alternate” is not the correct word to use and we have chosen to remove it, we only keep the word “damage” (p. 2, line 83).
Comment #7: Table 5. I question the need for this table. All treatments were in closed systems so the results should be the same for all samples.
Response: We understand, we have removed Table 5 and given the information in text on lines 306-308 instead.
Comment #8: Lines 393-395 in the Conclusions. This belongs in the discussion not in the Conclusions.
Response: The content has been moved to Section 3.3 (p. 10, lines 346-350) and the conclusion has been rewritten (p. 15, lines 426-427).

Round 2
Reviewer 2 Report
Comments and Suggestions for Authors
I am satisfied with the authors' explanation. I recommend relying on previous and relevant work. In addition, the units in L306 should be included.
Author Response
Dear Reviewer 2
Thank you for the valuable comments. We have done our best to make the suggested changes and hope you find them satisfactory in the revised manuscript. Please find our responses to the comments below. In the revised manuscript new and significantly modified sections are indicated with red font.
Comments and Suggestions for Authors
I am satisfied with the authors' explanation. I recommend relying on previous and relevant work. In addition, the units in L306 should be included.
Response: Thank you for your comment, the unit has been corrected (p. 9, line 297).
Reviewer 3 Report
Comments and Suggestions for Authors
The authors have satisfactorily addressed the comments in my review.